# Projecting Ising Model Parameters for Fast Mixing

**Justin Domke**
NICTA, The Australian National University
justin.domke@nicta.com.au

**Xianghang Liu**
NICTA, The University of New South Wales
xianghang.liu@nicta.com.au

## Abstract

Inference in general Ising models is difficult, due to high treewidth making tree-based algorithms intractable. Moreover, when interactions are strong, Gibbs sampling may take exponential time to converge to the stationary distribution. We present an algorithm to project Ising model parameters onto a parameter set that is guaranteed to be fast mixing, under several divergences. We find that Gibbs sampling using the projected parameters is more accurate than with the original parameters when interaction strengths are strong and when limited time is available for sampling.

## 1 Introduction

High-treewidth graphical models typically yield distributions where exact inference is intractable. To cope with this, one often makes an approximation based on a tractable model. For example, given some intractable distribution $q$, mean-field inference [14] attempts to minimize $KL(p||q)$ over $p \in \text{TRACT}$, where TRACT is the set of fully-factorized distributions. Similarly, structured mean-field minimizes the KL-divergence, but allows TRACT to be the set of distributions that obey some tree [16] or a non-overlapping clustered [20] structure. In different ways, loopy belief propagation [21] and tree-reweighted belief propagation [19] also make use of tree-based approximations, while Globerson and Jaakkola [6] provide an approximate inference method based on exact inference in planar graphs with zero field.

In this paper, we explore an alternative notion of a "tractable" model. These are "fast mixing" models, or distributions that, while they may be high-treewidth, have parameter-space conditions guaranteeing that Gibbs sampling will quickly converge to the stationary distribution. While the precise form of the parameter space conditions is slightly technical (Sections 2-3), informally, it is simply that interaction strengths between neighboring variables are not too strong.

In the context of the Ising model, we attempt to use these models in the most basic way possible– by taking an arbitrary (slow-mixing) set of parameters, projecting onto the fast-mixing set, using four different divergences. First, we show how to project in the Euclidean norm, by iteratively thresholding a singular value decomposition (Theorem 7). Secondly, we experiment with projecting using the "zero-avoiding" divergence $KL(p||q)$. Since this requires taking (intractable) expectations with respect to $q$, it is of only theoretical interest. Third, we suggest a novel "piecewise" approximation of the KL divergence, where one drops edges from both $q$ and $p$ until a low-treewidth graph remains where the exact KL divergence can be calculated. Experimentally, this does not perform as well as the true KL-divergence, but is easy to evaluate. Fourth, we consider the "zero forcing" divergence $KL(q||p)$. Since this requires expectations with respect to $p$, which is constrained to be fast-mixing, it can be approximated by Gibbs sampling, and the divergence can be minimized through stochastic approximation. This can be seen as a generalization of mean-field where the set of approximating distributions is expanded from fully-factorized to fast-mixing.

## 2 Background

The literature on mixing times in Markov chains is extensive, including a recent textbook [10]. The presentation in the rest of this section is based on that of Dyer et al. [4].

Given a distribution $p(x)$, one will often wish to draw samples from it. While in certain cases (e.g. the Normal distribution) one can obtain exact samples, for Markov random fields (MRFs), one must generally resort to iterative Markov chain Monte Carlo (MCMC) methods that obtain a sample asymptotically. In this paper, we consider the classic Gibbs sampling method [5], where one starts with some configuration $x$, and repeatedly picks a node $i$, and samples $x_i$ from $p(x_i|x_{-i})$. Under mild conditions, this can be shown to sample from a distribution that converges to $p$ as $t \to \infty$.

It is common to use more sophisticated methods such as block Gibbs sampling, the Swendsen-Wang algorithm [18], or tree sampling [7]. In principle, each algorithm could have unique parameter-space conditions under which it is fast mixing. Here, we focus on the univariate case for simplicity and because fast mixing of univariate Gibbs is sufficient for fast mixing of some other methods [13].

**Definition 1.** Given two finite distributions $p$ and $q$, the **total variation distance** $||\cdot||_{TV}$ is

$$||p(X) - q(X)||_{TV} = \frac{1}{2} \sum_x |p(X = x) - q(X = x)|.$$

We need a property of a distribution that can guarantee fast mixing. The **dependency** $R_{ij}$ of $x_i$ on $x_j$ is defined by considering two configurations $x$ and $x'$, and measuring how much the conditional distribution of $x_i$ can vary when $x_k = x'_k$ for all $k \neq j$.

**Definition 2.** Given a distribution $p$, the dependency matrix $R$ is defined by

$$R_{ij} = \max_{x,x':x_{-j}=x'_{-j}} ||p(X_i|x_{-i}) - p(X_i|x'_{-i})||_{TV}.$$

Given some threshold $\epsilon$, the **mixing time** is the number of iterations needed to guarantee that the total variation distance of the Gibbs chain to the stationary distribution is less than $\epsilon$.

**Definition 3.** Suppose that $\{X^t\}$ denotes the sequence of random variables corresponding to running Gibbs sampling on some distribution $p$. The mixing time $\tau(\epsilon)$ is the minimum time $t$ such that the total variation distance between $X^t$ and the stationary distribution is at most $\epsilon$. That is,

$$\tau(\epsilon) = \min\{t : d(t) < \epsilon\},$$
$$d(t) = \max_x ||\mathbb{P}(X^t|X^0 = x) - p(X)||_{TV}.$$

Unfortunately, the mixing time can be extremely long, which makes the use of Gibbs sampling delicate in practice. For example, for the two-dimensional Ising model with zero field and uniform interactions, it is known that mixing time is polynomial (in the size of the grid) when the interaction strengths are below a threshold $\beta_c$, and exponential for stronger interactions [11]. For more general distributions, such tight bounds are not generally known, but one can still derive sufficient conditions for fast mixing. The main result we will use is the following [8].

**Theorem 4.** *Consider the dependency matrix $R$ corresponding to some distribution $p(X_1, ..., X_n)$. For Gibbs sampling with random updates, if $||R||_2 < 1$, the mixing time is bounded by*

$$\tau(\epsilon) \leq \frac{n}{1 - ||R||_2} \ln\left(\frac{n}{\epsilon}\right).$$

Roughly speaking, if the spectral norm (maximum singular value) of $R$ is less than one, rapid mixing will occur. A similar result holds in the case of systematic scan updates [4, 8].

Some of the classic ways of establishing fast mixing can be seen as special cases of this. For example, the Dobrushin criterion is that $||R||_1 < 1$, which can be easier to verify in many cases, since $||R||_1 = \max_j \sum_i |R_{ij}|$ does not require the computation of singular values. However, for symmetric matrices, it can be shown that $||R||_2 \leq ||R||_1$, meaning the above result is tighter.

# 3 Mixing Time Bounds

For variables $x_i \in \{-1, +1\}$, an Ising model is of the form

$$p(x) = \exp\left(\sum_{i,j} \beta_{ij} x_i x_j + \sum_i \alpha_i x_i - A(\beta, \alpha)\right),$$

where $\beta_{ij}$ is the interaction strength between variables $i$ and $j$, $\alpha_i$ is the "field" for variable $i$, and $A$ ensures normalization. This can be seen as a member of the exponential family $p(x) = \exp\left(\theta \cdot f(x) - A(\theta)\right)$, where $f(x) = \{x_i x_j \forall (i,j)\} \cup \{x_i \forall i\}$ and $\theta$ contains both $\beta$ and $\alpha$.

**Lemma 5.** *For an Ising model, the dependency matrix is bounded by*
$$R_{ij} \leq \tanh |\beta_{ij}| \leq |\beta_{ij}|$$

Hayes [8] proves this for the case of constant $\beta$ and zero-field, but simple modifications to the proof can give this result.

Thus, to summarize, an Ising model can be guaranteed to be fast mixing if the spectral norm of the absolute value of interactions terms is less than one.

# 4 Projection

In this section, we imagine that we have some set of parameters $\theta$, not necessarily fast mixing, and would like to obtain another set of parameters $\psi$ which are as close as possible to $\theta$, but guaranteed to be fast mixing. This section derives a projection in the Euclidean norm, while Section 5 will build on this to consider other divergence measures.

We will use the following standard result that states that given a matrix $A$, the closest matrix with a maximum spectral norm can be obtained by thresholding the singular values.

**Theorem 6.** *If $A$ has a singular value decomposition $A = USV^T$, and $||\cdot||_F$ denotes the Frobenius norm, then $B = \arg\min_{B:||B||_2 \leq c} ||A - B||_F$ can be obtained as $B = US'V^T$, where $S'_{ii} = \min(S_{ii}, c^2)$.*

We denote this projection by $B = \Pi_c[A]$. This is close to providing an algorithm for obtaining the closest set of Ising model parameters that obey a given spectral norm constraint. However, there are two issues. First, in general, even if $A$ is sparse, the projected matrix $B$ will be dense, meaning that projecting will destroy a sparse graph structure. Second, this result constrains the spectral norm of $B$ itself, rather than $R = |B|$, which is what needs to be controlled. The theorem below provides a dual method that fixed these issues.

Here, we take some matrix $Z$ that corresponds to the graph structure, by setting $Z_{ij} = 0$ if $(i,j)$ is an edge, and $Z_{ij} = 1$ otherwise. Then, enforcing that $B$ obeys the graph structure is equivalent to enforcing that $Z_{ij} B_{ij} = 0$ for all $(i,j)$. Thus, finding the closest set of parameters $B$ is equivalent to solving

$$\min_{B,D} \quad ||A - B||_F \text{ subject to } ||D||_2 \leq c, \ Z_{ij} D_{ij} = 0, \ D = |B|. \tag{1}$$

We find it convenient to solve this minimization by performing some manipulations, and deriving a dual. The proof of this theorem is provided in the appendix. To accomplish the maximization of $g$ over $M$ and $\Lambda$, we use LBFGS-B [1], with bound constraints used to enforce that $M \geq 0$.

The following theorem uses the "triple dot product" notation of $A \cdot B \cdot C = \sum_{ij} A_{ij} B_{ij} C_{ij}$.

**Theorem 7.** *Define $R = |A|$. The minimization in Eq. 1 is equivalent to the problem of $\max_{M \geq 0, \Lambda} g(\Lambda, M)$, where the objective and gradient of $g$ are, for $D(\Lambda, M) = \Pi_c[R + M - \Lambda \odot Z]$,*

$$g(\Lambda, M) = \frac{1}{2}||D(\Lambda, M) - R||_F^2 + \Lambda \cdot Z \cdot D(\Lambda, M) \tag{2}$$

$$\frac{dg}{d\Lambda} = Z \odot D(\Lambda, M) \tag{3}$$

$$\frac{dg}{dM} = D(\Lambda, M). \tag{4}$$

# 5  Divergences

Again, we would like to find a parameter vector $\psi$ that is close to a given vector $\theta$, but is guaranteed to be fast mixing, but with several notions of "closeness" that vary in terms of accuracy and computational convenience. Formally, if $\Psi$ is the set of parameters that we can guarantee to be fast mixing, and $D(\theta, \psi)$ is a divergence between $\theta$ and $\psi$, then we would like to solve

$$\arg \min_{\psi \in \Psi} D(\theta, \psi). \tag{5}$$

As we will see, in selecting $D$ there appears to be something of a trade-off between the quality of the approximation, and the ease of computing the projection in Eq. 5.

In this section, we work with the generic exponential family representation

$$p(x; \theta) = \exp(\theta \cdot f(x) - A(\theta)).$$

We use $\mu$ to denote the mean value of $f$. By a standard result, this is equal to the gradient of $A$, i.e.

$$\mu(\theta) = \sum_x p(x; \theta) f(x) = \nabla A(\theta).$$

## 5.1  Euclidean Distance

The simplest divergence is simply the $l_2$ distance between the parameter vectors, $D(\theta, \psi) = ||\theta - \psi||_2$. For the Ising model, Theorem 7 provides a method to compute the projection $\arg \min_{\psi \in \Psi} ||\theta - \psi||_2$. While simple, this has no obvious probabilistic interpretation, and other divergences perform better in the experiments below.

However, it also forms the basis of our projected gradient descent strategy for computing the projection in Eq. 5 under more general divergences $D$. Specifically, we will do this by iterating

1. $\psi' \leftarrow \psi - \lambda \frac{d}{d\psi} D(\theta, \psi)$
2. $\psi \leftarrow \arg \min_{\psi \in \Psi} ||\psi' - \psi||_2$

for some step-size $\lambda$. In some cases, $dD/d\psi$ can be calculated exactly, and this is simply projected gradient descent. In other cases, one needs to estimate $dD/d\psi$ by sampling from $\psi$. As discussed below, we do this by maintaining a "pool" of samples. In each iteration, a few Markov chain steps are applied with the current parameters, and then the gradient is estimated using them. Since the gradients estimated at each time-step are dependent, this can be seen as an instance of Ergodic Mirror Descent [3]. This guarantees convergence if the number of Markov chain steps, and the step-size $\lambda$ are both functions of the total number of optimization iterations.

## 5.2  KL-Divergence

Perhaps the most natural divergence to use would be the "inclusive" KL-divergence

$$D(\theta, \psi) = KL(\theta || \psi) = \sum_x p(x; \theta) \log \frac{p(x; \theta)}{p(x; \psi)}. \tag{6}$$

This has the "zero-avoiding" property [12] that $\psi$ will tend to assign some probability to all configurations that $\theta$ assigns nonzero probability to. It is easy to show that the derivative is

$$\frac{dD(\theta, \psi)}{d\psi} = \mu(\psi) - \mu(\theta), \tag{7}$$

where $\mu_\theta = \mathbb{E}_\theta[f(X)]$. Unfortunately, this requires inference with respect to both the parameter vectors $\theta$ and $\psi$. Since $\psi$ will be enforced to be fast-mixing during optimization, one could approximate $\mu(\psi)$ by sampling. However, $\theta$ is presumed to be slow-mixing, making $\mu(\theta)$ difficult to compute. Thus, this divergence is only practical on low-treewidth "toy" graphs.

## 5.3 Piecewise KL-Divergences

Inspired by the piecewise likelihood [17] and likelihood approximations based on mixtures of trees [15], we seek tractable approximations of the KL-divergence based on tractable subgraphs. Our motivation is the the following: if $\theta$ and $\psi$ define the same distribution, then if a certain set of edges are removed from both, they should continue to define the same distribution[1]. Thus, given some graph $T$, we define the "projection" $\theta(T)$ onto the tree such by setting all edge parameters to zero if not part of $T$. Then, given a set of graphs $T$, the piecewise KL-divergence is

$$D(\theta, \psi) = \max_{T} KL(\theta(T)||\psi(T)).$$

Computing the derivative of this divergence is not hard– one simply computes the KL-divergence for each graph, and uses the gradient as in Eq. 7 for the maximizing graph.

There is some flexibility of selecting the graphs $T$. In the simplest case, one could simply select a set of trees (assuring that each edge is covered by one tree), which makes it easy to compute the KL-divergence on each tree using the sum-product algorithm. We will also experiment with selecting low-treewidth graphs, where exact inference can take place using the junction tree algorithm.

## 5.4 Reversed KL-Divergence

We also consider the "zero-forcing" KL-divergence

$$D(\theta, \psi) = KL(\psi||\theta) = \sum_{x} p(x; \psi) \log \frac{p(x; \psi)}{p(x; \theta)}.$$

**Theorem 8.** *The divergence $D(\theta, \psi) = KL(\psi||\theta)$ has the gradient*

$$\frac{d}{d\psi} D(\theta, \psi) = \sum_{x} p(x; \psi)(\psi - \theta) \cdot f(x) \left( f(x) - \mu(\psi) \right).$$

Arguably, using this divergence is inferior to the "zero-avoiding" KL-divergence. For example, since the parameters $\psi$ may fail to put significant probability at configurations where $\theta$ does, using importance sampling to reweight samples from $\psi$ to estimate expectations with respect to $\theta$ could have high variance Further, it can be non-convex with respect to $\psi$. Nevertheless, it often work well in practice. Minimizing this divergence under the constraint that the dependency matrix $R$ corresponding to $\psi$ have a limited spectral norm is closely related to naive mean-field, which can be seen as a degenerate case where one constrains $R$ to have zero norm.

This is easier to work with than the "zero-avoiding" KL-divergence in Eq. 6 since it involves taking expectations with respect to $\psi$, rather than $\theta$: since $\psi$ is enforced to be fast-mixing, these expectations can be approximated by sampling. Specifically, suppose that one has generated a set of samples $x^1, ..., x^K$ using the current parameters $\psi$. Then, one can first approximate the marginals by $\hat{\mu} = \frac{1}{K} \sum_{k=1}^{K} f(x^k)$, and then approximate the gradient by

$$\hat{g} = \frac{1}{K} \sum_{k=1}^{K} \left( (\psi - \theta) \cdot f(x^k) \right) \left( f(x^k) - \hat{\mu} \right). \tag{8}$$

It is a standard result that if two estimators are unbiased and independent, the product of the two estimators will also be unbiased. Thus, if one used separate sets of perfect samples to estimate $\hat{\mu}$ and $\hat{g}$, then $\hat{g}$ would be an unbiased estimator of $dD/d\psi$. In practice, of course, we generate the samples by Gibbs sampling, so they are not quite perfect. We find in practice that using the same set of samples twice makes makes little difference, and do so in the experiments.

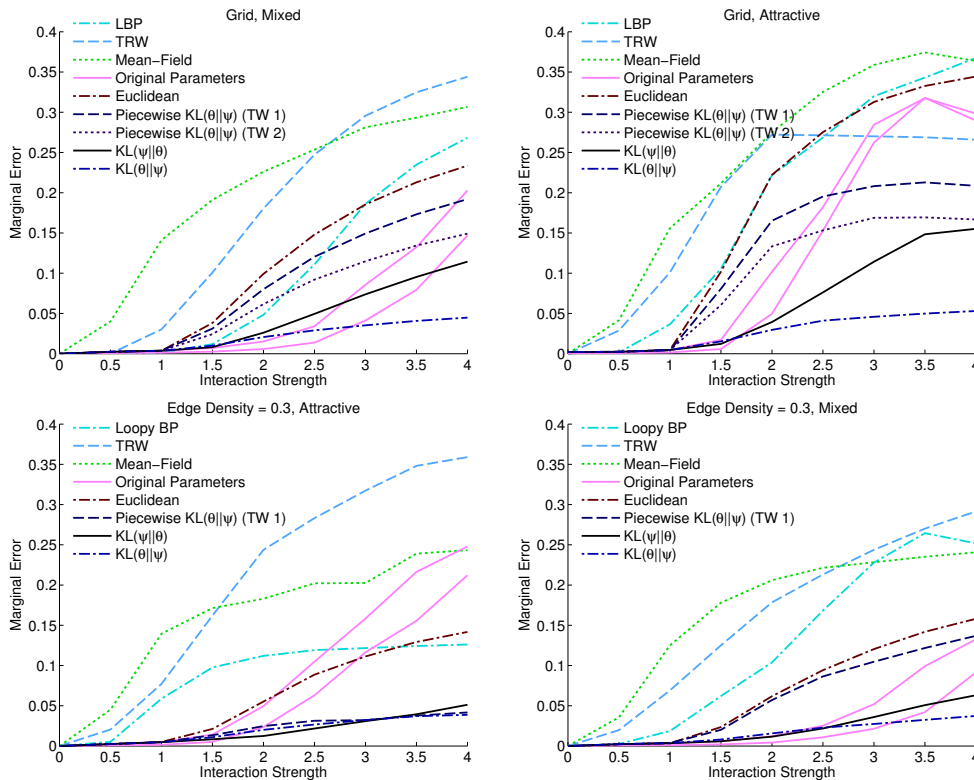

Figure 1: The mean error of estimated univariate marginals on 8x8 grids (top row) and low-density random graphs (bottom row), comparing 30k iterations of Gibbs sampling after projection to variational methods. To approximate the computational effort of projection (Table 1), sampling on the original parameters with 250k iterations is also included as a lower curve. (Full results in appendix.)

# 6 Experiments

Our experimental evaluation follows that of Hazan and Shashua [9] in evaluating the accuracy of the methods using the Ising model in various configurations. In the experiments, we approximate randomly generated Ising models with rapid-mixing distributions using the projection algorithms described previously. Then, the marginals of rapid-mixing approximate distribution are compared against those of the target distributions by running a Gibbs chain on each. We calculate the mean absolute distance of the marginals as the accuracy measure, with the marginals computed via the exact junction-tree algorithm.

We evaluate projecting under the Euclidean distance (Section 5.1), the piecewise divergence (Section 5.3), and the zero-forcing KL-divergence $KL(\psi||\theta)$ (Section 5.4). On small graphs, it is possible to minimize the zero-avoiding KL-divergence $KL(\theta||\psi)$ by computing marginals using the junction-tree algorithm. However, as minimizing this KL-divergence leads to exact marginal estimates, it doesn't provide a useful measure of marginal accuracy. Our methods are compared with four other inference algorithms, namely loopy belief-propagation (LBP), Tree-reweighted belief-propagation (TRW), Naive mean-field (MF), and Gibbs sampling on the original parameters.

LBP, MF and TRW are among the most widely applied variational methods for approximate inference. The MF algorithm uses a fully factorized distribution as the tractable family, and can be viewed as an extreme case of minimizing the zero forcing KL-divergence $KL(\psi||\theta)$ under the constraint of zero spectral norm. The tractable family that it uses guarantees "instant" mixing but is much more restrictive. Theoretically, Gibbs sampling on the original parameters will produce highly accurate marginals if run long enough. However, this can take exponentially long and convergence is generally hard to diagnose [2]. In contrast, Gibbs sampling on the rapid-mixing approximation is guaranteed to converge rapidly but will result in less accurate marginals asymptotically. Thus, we also include time-accuracy comparisons between these two strategies in the experiments.

| | Grid, Strength 1.5 | | Grid, Strength 3 | | Random Graph, Strength 3. | |
|---|---|---|---|---|---|---|
| | Gibbs Steps | SVDs | Gibbs Steps | SVDs | Gibbs Steps | SVDs |
| 30,000 Gibbs steps | 30k / 0.17s | | 30k / 0.17s | | 30k / 0.04s | |
| 250,000 Gibbs steps | 250k / 1.4s | | 250k / 1.4s | | 250k / 0.33s | |
| Euclidean Projection | | 22 / 0.04s | | 78 / 0.15s | | 17 / .0002s |
| Piecewise-1 Projection | | 322 / 0.61s | | 547 / 1.0s | | 408 / 0.047s |
| KL Projection | 30k / 0.17s | 265 / 0.55s | 30k / 0.17s | 471 / 0.94s | 30k / 0.04s | 300 / 0.037s |

Table 1: Running Times on various attractive graphs, showing the number of Gibbs passes and Singular Value Decompositions, as well as the amount of computation time. The random graph is based on an edge density of 0.7. Mean-Field, Loopy BP, and TRW take less than 0.01s.

## 6.1 Configurations

Two types of graph topologies are used: two-dimensional $8 \times 8$ grids and random graphs with $10$ nodes. Each edge is independently present with probability $p_e \in \{0.3, 0.5, 0.7\}$. Node parameters $\theta_i$ are uniformly drawn from unif$(-d_n, d_n)$ and we fix the field strength to $d_n = 1.0$. Edge parameters $\theta_{ij}$ are uniformly drawn from unif$(-d_e, d_e)$ or unif$(0, d_e)$ to obtain mixed or attractive interactions respectively. We generate graphs with different interaction strength $d_e = 0, 0.5, \ldots, 4$. All results are averaged over 50 random trials.

To calculate piecewise divergences, it remains to specify the set of subgraphs $T$. It can be any tractable subgraph of the original distribution. For the grids, one straightforward choice is to use the horizontal and the vertical chains as subgraphs. We also test with chains of treewidth 2. For random graphs, we use the sets of random spanning trees which can cover every edge of the original graphs as the set of subgraphs.

A stochastic gradient descent algorithm is applied to minimize the zero forcing KL-divergence $KL(\psi||\theta)$. In this algorithm, a "pool" of samples is repeatedly used to estimate gradients as in Eq. 8. After each parameter update, each sample is updated by a single Gibbs step, consisting of one pass over all variables. The performance of this algorithm can be affected by several parameters, including the gradient search step size, the size of the sample pool, the number of Gibbs updates, and the number of total iterations. (This algorithm can be seen as an instance of Ergodic Mirror Descent [3].) Without intensive tuning of these parameters, we choose a constant step size of $0.1$, sample pool size of 500 and 60 total iterations, which performed reasonably well in practice.

For each original or approximate distribution, a single chain of Gibbs sampling is run on the final parameters, and marginals are estimated from the samples drawn. Each Gibbs iteration is one pass of systematical scan over the variables in fixed order. Note that this does not take into account the computational effort deployed during projection, which ranges from 30,000 total Gibbs iterations with repeated Euclidean projection $(KL(\psi||\theta))$ to none at all (Original parameters). It has been our experience that more aggressive parameters can lead to this procedure being more accurate than Gibbs in a comparison of total computational effort, but such a scheduling tends to also reduce the accuracy of the final parameters, making results more difficult to interpret.

In Section 3.2, we show that for Ising models, a sufficient condition for rapid-mixing is the spectral norm of pairwise weight matrix is less than 1.0. However, we find in practice using a spectral norm bound of 2.5 instead of 1.0 can still preserve the rapid-mixing property and gives better approximation to the original distributions. (See Section 7 for a discussion.)

## 7 Discussion

Inference in high-treewidth graphical models is intractable, which has motivated several classes of approximations based on tractable families. In this paper, we have proposed a new notion of "tractability", insisting not that a graph has a fast algorithm for exact inference, but only that it obeys parameter-space conditions ensuring that Gibbs sampling will converge rapidly to the stationary distribution. For the case of Ising models, we use a simple condition that can guarantee rapid mixing, namely that the spectral norm of the matrix of interaction strengths is less than one.

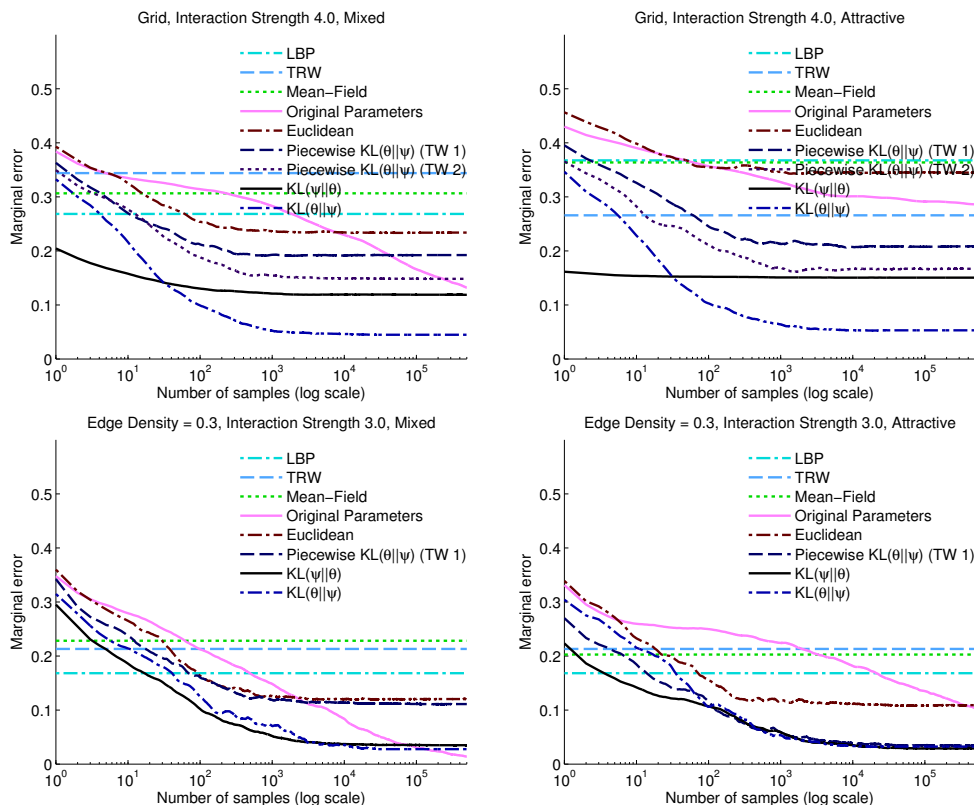

Figure 2: Example plots of the accuracy of obtained marginals vs. the number of samples. Top: Grid graphs. Bottom: Low-Density Random graphs. (Full results in appendix.)

Given an intractable set of parameters, we consider using this approximate family by "projecting" the intractable distribution onto it under several divergences. First, we consider the Euclidean distance of parameters, and derive a dual algorithm to solve the projection, based on an iterative thresholding of the singular value decomposition. Next, we extend this to more probabilistic divergences. Firstly, we consider a novel "piecewise" divergence, based on computing the exact KL-divergnce on several low-treewidth subgraphs. Secondly, we consider projecting onto the KL-divergence. This requires a stochastic approximation approach where one repeatedly generates samples from the model, and projects in the Euclidean norm after taking a gradient step.

We compare experimentally to Gibbs sampling on the original parameters, along with several standard variational methods. The proposed methods are more accurate than variational approximations. Given enough time, Gibbs sampling using the original parameters will always be more accurate, but with finite time, projecting onto the fast-mixing set to generally gives better results.

Future work might extend this approach to general Markov random fields. This will require two technical challenges. First, one must find a bound on the dependency matrix for general MRFs, and secondly, an algorithm is needed to project onto the fast-mixing set defined by this bound. Fast-mixing distributions might also be used for learning. E.g., if one is doing maximum likelihood learning using MCMC to estimate the likelihood gradient, it would be natural to constrain the parameters to a fast mixing set.

One weakness of the proposed approach is the apparent looseness of the spectral norm bound. For the two dimensional Ising model with no univariate terms, and a constant interaction strength $\beta$, there is a well-known threshold $\beta_c = \frac{1}{2}\ln(1 + \sqrt{2}) \approx .4407$, obtained using more advanced techniques than the spectral norm [11]. Roughly, for $\beta < \beta_c$, mixing is known to occur quickly (polynomial in the grid size) while for $\beta > \beta_c$, mixing is exponential. On the other hand, the spectral bound norm will be equal to one for $\beta = .25$, meaning the bound is too conservative in this case by a factor of $\beta_c/.25 \approx 1.76$. A tighter bound on when rapid mixing will occur would be more informative.

## Footnotes

[1]Technically, here, we assume that the exponential family is minimal. However, in the case of an over-complete exponential family, enforcing this will simply ensure that $\theta$ and $\psi$ use the same reparameterization.

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
