[Supplementary Material · ProjectingIsing_appendix.pdf]

## Appendix

Recall that we are interested in the minimization

$$\min_{B,D} \quad ||A - B||_F \tag{9}$$
$$s.t. \quad ||D||_2 \leq c$$
$$Z_{ij}D_{ij} = 0$$
$$D = |B|.$$

**Lemma 9.** *If we define $R = |A|$, this is equivalent to the minimization*

$$\min_{D} \quad ||R - D||_F \tag{10}$$
$$s.t. \quad ||D||_2 \leq c$$
$$Z_{ij}D_{ij} = 0$$
$$D \geq 0$$

*Proof.* For fixed $D$, the minimum $B$ will always be achieved by $B = D \odot \text{sign}(A)$, meaning $||A - B||_F = ||A - D \odot \text{sign}(A)||_F = ||R - D||_F$. □

To actually project the parameters $A = (\beta_{ij})$ corresponding to an Ising model, one first takes the absolute value $R = |A|$, and passes it as input to this minimization. After finding the minimizing argument, the new parameters are $B = D \odot \text{sign}(A)$.

**Theorem 10.** *Define $R = |A|$. The minimization in Eq. 1 is equivalent to the problem of $\max_{M \geq 0, \Lambda} g(\Lambda, M)$, where the objective and gradient of $g$ are, for $D(\Lambda, M) = \Pi_c[R + M - \Lambda \odot Z]$,*

$$g(\Lambda, M) = \frac{1}{2}||D(\Lambda, M) - R||_F^2 + \Lambda \cdot Z \cdot D(\Lambda, M) \tag{11}$$

$$\frac{dg}{d\Lambda} = Z \odot D(\Lambda, M) \tag{12}$$

$$\frac{dg}{dM} = D(\Lambda, M). \tag{13}$$

*Proof.* The minimization in Eq. 10 has the Lagrangian

$$\mathcal{L}(D, \Lambda, M) = \frac{1}{2}||D - R||_F^2 + I[||D||_2 \leq c] + \Lambda \cdot Z \cdot D - M \cdot D, \tag{14}$$

where $I$ is an indicator function returning $\infty$ if $||D||_2 > c$ and zero otherwise, $\Lambda$ is a matrix of Lagrange multipliers enforcing that $Z_{ij}D_{ij} = 0$, and $M$ is a matrix of Lagrange multipliers enforcing that $D \geq 0$.

Standard duality theory states that Eq. 10 is equivalent to the saddle-point problem $\max_{M \geq 0, \Lambda} \min_D \mathcal{L}(D, \Lambda, M)$. So, we are interested in evaluating $g(\Lambda, M) = \min_D \mathcal{L}(D, \Lambda, M)$ for fixed $\Lambda$ and $M$. Some algebra gives

$$\arg\min_D \mathcal{L}(D, \Lambda, M)$$

$$= \arg\min_D \frac{1}{2}||D - R||_F^2 + \Lambda \cdot Z \cdot D + I[||D||_2 \leq c] - M \cdot D$$

$$= \arg\min_D \frac{1}{2}||D - (R + M - \Lambda \odot Z)||_F^2 + I[||D||_2 \leq c],$$

which shows that $g$ can be calculated as in Eq. 11.

Next, we are interested in the gradient of $g$. By applying Danskin's theorem to Eq. 14, we have that $\frac{d}{dM} \arg\min_D \mathcal{L}(D, \Lambda, M)$ will be exactly $D$ where $D$ is the minimizer of Eq. 14. This establishes Eq. 13. Similarly, it can be shown that $\frac{d}{d\Lambda} \arg\min_D \mathcal{L}(D, \Lambda, M) = Z \odot D$, establishing Eq. 12. □

Figure 3: Accuracy on Grids, as a function of edge strength. All sampling methods use 30k samples, except sampling on the original parameters which includes a second (lower) curve with 250k samples.

**Theorem 11.** *The divergence $D(\theta, \psi) = KL(\psi||\theta)$ has the gradient*

$$\frac{d}{d\psi}D(\theta, \psi) = \sum_x p(x; \psi)(\psi - \theta) \cdot f(x)\left(f(x) - \mu(\psi)\right).$$

*Proof.* Firstly, it can be shown that

$$D(\theta, \psi) = \sum_x p(x; \psi)(\psi - \theta) \cdot f(x) + A(\theta) - A(\psi).$$

From this, it follows that

$$\frac{d}{d\psi}D(\theta, \psi) = \sum_x \frac{dp(x; \psi)}{d\psi}(\psi - \theta) \cdot f(x)$$
$$+ \sum_x p(x; \psi)f(x) - \mu(\psi).$$

This can be seen to be equivalent to the result by observing that the second two terms cancel, and that $dp(x; \psi)/d\psi = p(x; \psi)(f(x) - \mu(\psi))$. □

Figure 4: Accuracy on random graphs, as a function of edge strength. All sampling methods use 30k samples, except sampling on the original parameters which includes a second (lower) curve with 250k samples.

Figure 5: Accuracy on Grids as a function of time

Figure 6: Accuracy on Low-Density Graphs as a function of time

Figure 7: Accuracy on Medium-Density Graphs as a function of time

Figure 8: Accuracy on High-Density Random Graphs as a function of time