[Reviews · NeurIPS 2013]

Submitted by Assigned_Reviewer_9

Aims to improve the mixing rate of Gibbs sampling in pairwise Ising models with strong interactions, which are known to be "slow-mixing". Several projections to "fast-mixing" models are proposed; essentially, parameters are identified which are as close as possible to the original model, but weak enough to satisfy a spectral bound establishing rapid mixing. Experiments show some regimes where this leads to improved marginal estimates for a given computation time, compared to variational methods (mean field and belief propagation variants) and Gibbs sampling in the original model.

The technical content builds heavily on prior results establishing conditions for rapid mixing [4,8]. But I haven't seen the idea of projecting onto the nearest rapidly mixing model, as an approximate inference method, explored before.

On the domain of "toy tiny Ising models", results offer some nice improvements over a good set of baselines (standard sampling and contemporary variational methods). My main concern is that I think there should be more than the usual amount of skepticism as to whether these results will scale to larger models. Experiments involve comparisons of mixing times of samplers on different models, and it is hard to judge how these will scale with problem size. Also there is a (from all appearances) computationally expensive projection step required to build the fast-mixing model, the cost of which seems not to be accounted for. Finally, it is not at all clear that generalization beyond binary states will be possible, since establishing convergence bounds more generally is far more challenging.

CLARITY:
In general the presentation is clear and reasonably accessible, given the technical content. But there are some places where reorganization and clarification is needed:
* The paper refers several times to "standard results" and even states these in the form of a theorem (e.g. Theorem 6) without reference or proof sketch. Lemma 5 is left unproven, the reference proves for a special case of zero-field.
* Discuss properties of the dependency matrix R. For instance, it does not appear to be tractable due to the maximization, please state whether this is the case. (I presume this is why lemma 5 is invoked.)
* The second half of Sec. 4 is nearly impossible to follow. Before Theorem 7 the text references g, M, and Lambda before they are introduced. Then the statement of Theorem 4 includes notation that is not really explained. If this optimization is going to be discussed, more explanation is needed. (Perhaps less time could be spent restating results from [8] which are not really used.)
* In Sec. 1, KL(q||p) is used for both directions of KL divergence when the notation needs to be shifted for one. In the first paragraph, q is used for the true distribution and p for the tractable approximation; this is the opposite of almost all related literature.

EXPERIMENTS:
It appears that the time comparison in Figure 2 does *not* include the computation required for projection. A somewhat ambiguous statement to this effect appears in Sec 6.1 but is unclear; please clarify and if it is the case, show results with projection time included. As it stands, the proposed methods essentially get to use the output of a sophisticated variational optimization without penalty, which certainly makes the improvement over standard Gibbs less convincing.

It was disappointing not to see experiments on larger models, given that Gibbs mixing times often depend on problem size. There are certainly options for running experiments in regimes where junction tree is intractable, like using a model where symmetries let the true marginals be computed, or taking the output of a very long sampling run as truth.

I understand that KL(\theta||\psi) is intractable in general, but it would still be interesting to explore here as a potential "best case" for how sampling in an approximate model would perform. (Junction tree could be used for the toy models used in the submission.)

Mean field is a degenerate case of the reverse KL projection, as the paper points out, yet there is a large difference between mean field error and the error from reverse KL projection. This deserves discussion.
Summary: The idea of approximating slow-mixing models by projecting to the closest fast-mixing model is a nice one, and recent work on mixing bounds is leveraged in an elegant way. But there are some concerns about experimental comparisons, and the limited range of models to which this approach is potentially applicable.

Submitted by Assigned_Reviewer_11

The authors propose a method for improving the mixing rate of Gibbs sampling in Ising models by projecting the models original parameterization onto a new parameter setting that satisfies the Dobrushin criterion needed for fast mixing. They formulate the projection as a constrained optimization problem, where the constraints are needed to ensure that the new parameters are defined over the same space as the original parameters, and examine this projection under several distance/divergence measures.

In my opinion, methods that combine principles from stochastic and deterministic inference is an under-explored area and as a result, I think this is an interesting idea. While the idea of augmenting the original parameters to improve mixing time is straightforward, I found the description of the dual of the projection problem to be a little unclear - e.g. how do the z_ij*d_ij =0 constraints ensure that the new parameterization is over the same space (can't it be over a smaller space)? I also was unsure of the overall procedure - do you perform the projected gradient operations to completion and then run a Gibbs sampler, or do you somehow interleave sampling with the gradient updates? An algorithm description box would help to clarify. Also, is the proposed projected gradient descent strategy guaranteed to converge?

I also found the experiments to be a little unconvincing. In the first set of experiments, why was the Gibbs sampler run for 30K iterations? Since you are comparing a sampling method to deterministic methods, a comparison on the basis of time would seem more fair. Also, where are the error bars on these charts - the reader cannot tell if these results are significant. The second set of results compare the original/naive gibbs sampler with gibbs samplers under different projections and show that the projection does lead to faster mixing. However, it takes time to performing the projection operation and this is not accounted for in this comparison (e.g. if the projection takes 1 minute, and the naive Gibbs sampler can generate 10k samples in that time, then the projection might not be worthwhile). Last, why was there no comparison to blocked Gibbs samplers or Gogate's "Lifted Gibbs Sampler"?
Summary: All in all, a very nice idea. However, the development of the projection problem and proposed method could use a little work and a bolstered set of experiments are needed to convince me of the utility of the method.

Submitted by Assigned_Reviewer_13

Success of inference by Gibbs sampling in MRF (here, only with two
labels, ie, ising model) depends strongly on the mixing rate of the
underlying Monte Carlo Markov chain. The paper suggests the following
approach to inference:

1) Project the input model (which possibly does not mix fast) on the set
of models that do mix fast.

2) Do inference on the obtained model that mixes fast.

The space of fast-mixing models is defined by bounding the spectral
norm of the matrix of absolute values of Ising edge strengths.
"Projection" is defined by divergences of Gibbs distribution. It is
forced to preserve the graph structure. Projection in Euclid distance
is obtained by dualizing the initial task and using LBFGS-B. For
other divergences (KL, piecewise KL, and reversed KL divergences are
implemented), projected gradient algorithm is used. In reversed KL
divergence, Gibbs sampling (but on a fast mixing model) must be done
to compute the projection.

Extensive experiments on small random models are presented compare the
approximated marginals with the true marginals. The methods tested are
the proposed one (with all the above divergences) and loopy BP, TRW,
MF and Gibbs sampling on the original model. Not only accuracy but also
runtime-vs-accuracy evaluation is done. The experiments show that the
proposed methods consistently outperform TRW, MF and LBP in accuracy,
and for reasonable range of runtimes also Gibbs sampling on the
original model. Of the proposed methods, the one with reversed KL
divergence performs consistently best.

Comments:

The projected gradient algorithm from section 5.1 n fact has two
nested loops, the inner loop being LBFGS-B. Pls give details on when
the inner iterations are stopped.

It is not clear what the horizontal axis in the plots in Figure 2 (and
the supplement) means. It is titled "number of samples" but sampling
is used only for reversed KL divergence. I believe the horizontal
axis should be runtime of the algorithm. Similarly, why not to report
also runtime of LBP, TRW and MF. This would ensure fair
accuracy-runtime comparison of all tested algorithms. Please, clarify
this issue - without that it is hard to interpret the experiments. Give absolute running time in seconds.

Please consider experimental comparison with larger models. An interesting option is to use models from the paper

[Boris Flach: A class of random fields on complete graphs with tractable partition function,
to appear in TPAMI, available online]

which allow polynomial inference.

222: replace "onto the tree" with "onto graph T"

226: Shouldn't we ssy "subgradient" rather than "derivative"?
Summary: Interesting paper, convincing empirical results. Practical utility can be limited though due to high runtime (this needs clarification in rebuttal).
Author Feedback

Author rebuttal: Thanks to all the reviewers for helpful comments. We have a couple of minor comments for each reviewer, and a longer discussion of the efficiency of projection, which all reviewers asked about.)

(Minor Comments)

Reviewer_11

The z_ij*d_ij=0 constraints are to ensure that the new parameterization is not over a *larger* space.  Without these constraints, projection of a parameters on a graph would yield a densely connected graph.

Convergence is guaranteed, in a certain sense (see reference 3 on ergodic mirror descent).  However, as with mean-field, a non-convex objective is being fit, meaning local minima are possible.

Lifted Gibbs Sampling doesn't seem applicable here since we aren't in a Makov Logic Network setting, but one could certainly apply Blocked Gibbs sampling. As we note in the paper, the mixing time bound used here is sufficient to guarantee fast mixing for block Gibbs. (Though it could be loose.) We agree that extension of the mixing-time bounds and projections to these more complex samplers is important, but also quite challenging.

Reviewer_13

We have done some experiments comparing to large (e.g. 40x40) zero-field planar Ising models, which has results similar to those shown. (Although one must measure pairwise marginal accuracy, since true univariate marginals are always exactly uniform.) Thank you for the reference on regular fully-connected graphs. Along with attractive zero-field planar Ising models, this provides a very interesting class of large-scale models with tractable exact marginals.

Reviewer_9

It is indeed non-obvious how to generalize these results beyond the Ising model. We quickly outline how this might be done on lines 418-420. After writing this paper, we spent several months working on this generalization, and found that it is possible to do these types of projections with general MRFs, although the more general projection algorithm is considerably more complex.

We actually ran all the experiments including projection under KL(\theta||\psi) (with marginals computed via the junction tree algorithm) but removed it out of a concern it might be confusing, since it is intractable on large graphs. Still, as you'd expect, it does perform better than the "reversed" KL divergence, so we'd be happy to put the results back in.

We were also at first surprised by the difference of performance between mean-field and reverse KL divergence. As one increases the allowed spectral norm from zero (mean field) to 2.5, the quality of results smoothly interpolates, confirming the performance difference is just due to optimizing the divergence over a larger constraint set. Presumably, this continues as the norm increases further, but of course Gibbs sampling could be exponentially slow.

(Efficiency of Projection)

The experiments do not currently include the computational effort deployed to find the projected parameters (also mentioned on lines 354-358). Particularly with the SGD algorithm, finding the most efficient optimization parameters (sample pool size, # Markov transitions, LBFGS convergence threshold, gradient step size) is challenging. We did not attempt to find the fastest parameters, instead opting for conservative (slow, reliable) parameters to ensure transparency of the results. Our thinking was, first, the main idea is to show that strong fast-mixing approximations exist, and second that projection could be deployed "offline". (e.g. one might project once, and then do inference after conditioning on various inputs.)

That being said, in hindsight it seems obvious that readers would also be interested in the efficiency of projection as compared to Gibbs or variational methods, and the paper should certainly make such comparisons clear. Measurement is somewhat complicated by unoptimized implementation in an interpreted language, but the computational bottlenecks are (a) Gibbs sampling, and (b) Singular Value Decomposition, as called when optimizing the dual in Theorem 7. As these are implemented efficiently in C/Fortran, we report their times below for the 8x8 grid with attractive strength 3 interactions.

30,000 iterations of Gibbs sampling: 0.18s
A single SVD of a dependency matrix: 0.0026s
Euclidean projection: (93 SVDs): 0.15s
Piecewise-1 projection: (399 SVDs): 1.1s
KL via Stochastic Gradient Descent: (540 SVDs + 30,000 Gibbs steps): 1.4s+0.2 = 1.6s
Variational Methods: less than 0.001s

Here are the same measurements for strength 1.5 interactions:

Euclidean projection: (23 SVDs): 0.061s
Piecewise-1 projection: (206 SVDs): 0.51s
KL via Stochastic Gradient Descent: (252 SVDs + 30,000 Gibbs steps): 0.7s+0.2 = 0.9s
Variational Methods: less than 0.001s

Note here that, though SGD uses 60 Euclidean projections, the cost is much less than 60x as much, due to the use of warm-start.

Roughly speaking: with strength 3, a single Euclidean costs the same as 30,000 iterations of Gibbs, and KL projection costs the same as 300,000 iterations of Gibbs. With strength 1.5, Euclidean costs as 10,000 iterations of Gibbs, and KL costs the same as 30,000 iterations. The paper should definitely include a table of such timing measurements, and run Gibbs for 300,000 iterations (one more order of magnitude) to be equal to the cost of KL projection. Note that Gibbs will still clearly be inferior to projecting with strong interaction strengths and/or dense graphs. (Observe the essentially horizontal lines for original parameters in Figs 5-8.) A online hybrid sampling/projection algorithm would presumably be superior in this setting, but we believe this goes beyond what can be done in one paper.